# LEARNING DATA-DERIVED PRIVACY PRESERVING REPRESENTATIONS FROM INFORMATION METRICS

## ABSTRACT

It is clear that users should own and control their data and privacy. Utility providers are also becoming more interested in guaranteeing data privacy. Therefore, users and providers can and should collaborate in privacy protecting challenges, and this paper addresses this new paradigm. We propose a framework where the user controls what characteristics of the data they want to share (utility) and what they want to keep private (secret), without necessarily asking the utility provider to change its existing machine learning algorithms. We first analyze the space of privacy-preserving representations and derive natural information-theoretic bounds on the utility-privacy trade-off when disclosing a sanitized version of the data $X$. We present explicit learning architectures to learn privacy-preserving representations that approach this bound in a data-driven fashion. We describe important use-case scenarios where the utility providers are willing to collaborate with the sanitization process. We study *space-preserving* transformations where the utility provider can use the same algorithm on original and sanitized data, a critical and novel attribute to help service providers accommodate varying privacy requirements with a single set of utility algorithms. We illustrate this framework through the implementation of three use cases; *subject-within-subject*, where we tackle the problem of having a face identity detector that works only on a consenting subset of users, an important application, for example, for mobile devices activated by face recognition; *gender-and-subject*, where we preserve facial verification while hiding the gender attribute for users who choose to do so; and *emotion-and-gender*, where we hide independent variables, as is the case of hiding gender while preserving emotion detection.

## 1 INTRODUCTION, CHALLENGES, AND CONTRIBUTIONS

Individuals are sharing vast amounts of data on a daily basis; at the same time, advances in machine learning mean that service providers that receive this user data can infer increasingly more sensitive attributes about their users. Care must be taken to provide an appropriate protection for users in order to adhere to various privacy, legal, and ethical constraints. Critical to this is the capability of finding effective privacy-preserving data representations in a data-driven manner that are able to accommodate each user's individual needs. It is also in the interest of the utility provider to collaborate in this endeavor, the privacy-preserving data representation should account for data processing pipelines already in place. It would be impractical to tackle many of the privacy-preserving challenges today using model-based approaches, a data-driven approach is essential in allowing privatization mechanisms to keep up with new applications.

Each user should have the ability to define, sensitive and non-sensitive information associated with their data; we propose that a user and the service provider *collaborate* towards achieving *user-specific* privacy. This results in a system where the user sanitizes the data, at the sensing and/or transmitting stage without affecting the utility they need; at the same time, the provider can use the same processing pipeline for both sanitized and non-sanitized data. This is essential to ensure the service provider can accommodate several privacy requests with a single set of utility algorithms.

An example addressed by the proposed framework is to block, even at the sensor level, a device from constantly "listening." For example, mobile devices scan images until they detect the owner, and then they open. But all images not from the owner should be private. The proposed framework

will make such devices not to understand the data until the visual or sound trigger is detected, with the capability to do this at the sensor level and without modifying the existing recognition system.

This new paradigm of collaborative privacy environment is critical since it has also been shown that algorithmic or data augmentation and unpredictable correlations can break privacy Israel et al. (2014); Narayanan & Shmatikov (2008); Oh et al. (2016); Reuben et al. (2016). The impossibility of universal privacy protection has been studied extensively in the domain of differential privacy Dwork (2008), where a number of authors have shown that assumptions about the data or the adversary must be made in order to be able to provide utility Dwork & Naor (2010); Hardt et al. (2016); Kifer & Machanavajjhala (2011; 2014). We can, however, minimize the amount of privacy we are willing to sacrifice for a given level of utility. Other recent data-driven privacy approaches like Wu et al. (2018) have also explored this notion, but do not integrate the additional collaborative constraints.

Therefore, it is important to design collaborative systems where each user shares a sanitized version of their data with the service provider in such a way that user-defined non-sensitive tasks can be performed but user-defined sensitive ones cannot, without the service provider requiring to change any data processing pipeline otherwise.

**Contributions-** We consider a scenario where a user wants to share a sanitized representation of data $X$ in a way that a latent variable $U$ can be inferred, but a sensitive latent variable $S$ remains hidden. We formalize this notion using privacy and transparency definitions. We derive an information-theoretic bound on privacy-preserving representations. The metrics induced by this bound are used to learn such a representation directly from data, without prior knowledge of the joint distribution of the observed data $X$ and the latent variables $U$ and $S$. This process can accommodate for several user-specific privacy requirements, and can be modified to incorporate constraints about the service provider's existing utility inference algorithms enabling several privacy constraints to be satisfied in parallel for a given utility task.

We apply this framework to challenging use cases such as hiding gender information from a facial image (a relatively easy task) while preserving subject verification (a much harder task), or designing a sanitization function that preserves subject identification on a consenting subset of users, while disallowing it on the general population. Blocking a simpler task while preserving a harder one and blocking a device from constantly listening out-of-sample data are new applications in this work, here addressed with theoretical foundations and respecting the provider's existing algorithms, which can simultaneously handle sanitized and non-sanitized data.

The problem statement is detailed in Section 2, and the information-theoretic bounds are derived in Section 3. Section 4 defines a trainable adversarial game that directly attempts to achieve this bound; the section also discusses how service-provider specific requirements can be incorporated. Examples of this framework are shown in Section 5. The paper is concluded in Section 6. Complementary information and proofs are presented in the Supplementary Material.

## 2 PROBLEM STATEMENT

We describe a scenario in which we have access to possibly high-dimensional data $X \in \mathcal{X}$, this data depends on two special latent variables $U$ and $S$. $U$ is called the utility latent variable, and is a variable we want to communicate, while $S$ is called the secret, and is a variable we want to protect. We consider two agents, a service provider that wants to estimate $U$ from $X$, and an actor that wants to infer $S$ from $X$.

We define a third agent, the privatizer, that wants to learn a space-preserving stochastic mapping $Q : \mathcal{X} \to \mathcal{Q} \supset \mathcal{X}$ in such a way that $Q(X)$ provides information about the latent variable $U$, but provides relatively little information of $S$. In other words, we want to find a data representation that is private with respect to $S$ and transparent with respect to $U$.[1] We first recall the definition of privacy presented in Kifer (2009):

**Definition 2.1.** Privacy: Let $\delta_s$ be a measure of distance between probability distributions, $b_s \in \mathcal{R}^+$ a positive real number, and $P(S)$ the marginal distribution of the sensitive attribute $S$. The stochastic mapping $Q(X)$ is $(\delta_s, b_s)$-private with respect to $S$ if $\delta_s(P(S), P(S|Q(X))) < b_s$.

---

[1]Note that stochastic mappings are a natural extension of deterministic sanitization mappings. Furthermore, a deterministic mapping only truly destroys information if it is non-invertible.

We can define transparency in the same fashion:

**Definition 2.2.** Transparency: Let $\delta_u$ be a measure of distance between probability distributions, $b_u \in \mathcal{R}^+$ a positive real number, and $P(U|X)$ the posterior conditional distribution of the utility variable $U$ after observing $X$. The stochastic mapping $Q(X)$ is $(\delta_u, b_u)$-transparent with respect to $U$ if $\delta_u(P(U|X), P(U|Q(X))) < b_u$.

Both definitions depend on the learned mapping $Q$; in the following section, we derive an information-theoretic bound between privacy and transparency, and show that this bound infers a particular choice of metrics $\delta_u, \delta_s$. We then show that this inferred metric can be directly implemented as a loss function to learn privatization transformations from data using standard machine learning tools.

A similar analysis of these bounds for the special case where we directly observe the utility variable $U$ ($X = U$) was analyzed in Calmon et al. (2015) in the context of the Privacy Funnel. Here, we extend this to the more general case where $U$ is observed indirectly. More importantly, these bounds are used to design a data-driven implementation for learning privacy-preserving mappings.

# 3 INFORMATION-THEORETIC BOUNDS ON PRIVACY

Consider the utility and secret variables $U$ and $S$ defined over discrete alphabets $\mathcal{U}, \mathcal{S}$, and the observed data variable $X$, defined over $\mathcal{X}$, with joint distribution $P_{X,U,S}$. Figure 1a illustrates this set-up, and shows the fundamental relationship of their entropies $H(\cdot)$ and mutual information.

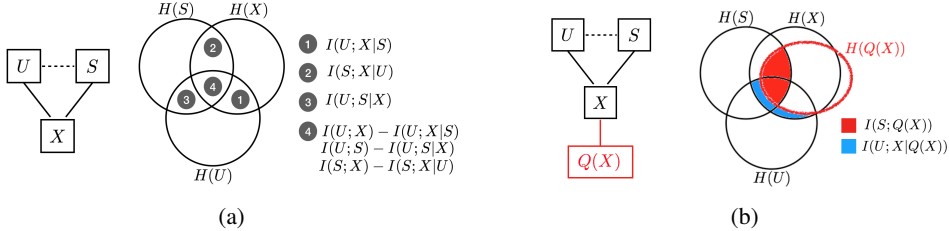

Figure 1: Left of figure (a) shows the dependency graph of the observed variable $X$ and the latent utility and secret variables $U$ and $S$. Right of figure (a) shows a Venn diagram illustrating conditional mutual informations $I$ that provide constraints on the performance of any sanitization mapping $Q(X)$. Left of figure (b) extends the dependency graph to show the sanitzed data $Q(X)$ in red. Right of figure (b) shows that the information leakage $I(S; Q(X))$ and censured information $I(U; X \mid Q(X))$ shown in red and blue respectively cannot be simultaneously set to 0, since they are partially at odds.

We analyze the properties of any mapping $Q : \mathcal{X} \to \mathcal{Q}$, and measure the resulting mutual information between the transformed variable $Q(X)$ and our quantities of interest. Our goal is to find $Q$ such that the information leakage from our sanitized data $I(S; Q(X))$ is minimized, while maximizing the shared information of the utility variable $I(U; Q(X))$. We will later relate these quantities and the bounds here developed with the privacy/utility definitions presented in the previous section.

Maximizing $I(U; Q(X))$ is equivalent to minimizing $I(U; X \mid Q(X))$, since $I(U; X|Q(X)) = I(U; X) - I(U; Q(X))$. The quantity $I(U; X \mid Q(X))$ is the information $X$ contains about $U$ that is censured by the sanitization mapping $Q$.

Figure 1b illustrates $I(S; Q(X))$ and $I(U; X|Q(X))$. One can see that there exists a trade-off area, $I(U, S) - I(U, S|X)$, that is always included in the union of $I(S; Q(X))$ and $I(U; X|Q(X))$. The lower we make $I(S; Q(X))$, the higher we make the censored information $I(U; X|Q(X))$, and vice versa. This induces a lower bound over the performance of the best possible mappings $Q(X)$ that is formalized in the following lemma.

*Lemma* 3.1. Let $X, U, S$ be three discrete random variables with joint probability distribution $P_{X,U,S}$. For any stochastic mapping $Q : \mathcal{X} \to \mathcal{Q}$ we have

$$I(U; S) - I(U; S|X) \leq [I(S; Q(X)] + [I(U; X|Q(X))]. \tag{1}$$

Proof of this lemma is shown in Supplementary Material. To show that this bound is reachable in some instances, consider the following example. Let $U$ and $S$ be independent discrete random variables, and $X = (U, S)$. The sanitization mapping $Q(X) = U$ satisfies this bound with equality.

We can also prove, trivially, an upper bound for these quantities.

*Lemma* 3.2. Let $X, U, S$ be three discrete random variables with joint probability distribution $P_{X,U,S}$. For any stochastic mapping $Q : \mathcal{X} \to \mathcal{Q}$ we have:

$$I(S; Q(X)) + I(U; X|Q(X)) \leq I(X; U, S). \tag{2}$$

That simply states that the information leakage about the secret and the censured information on the utility variable cannot exceed the total information present in the original observed variable $X$.

## 3.1 RELATION WITH PRIVACY AND TRANSPARENCY METRICS

We relate the terms $I(S; Q(X))$ and $I(U; X \mid Q(X))$ in Eq.1 back to our definitions of privacy and transparency.

$$
\begin{aligned}
I(U; X \mid Q) &= \sum_{q,x} p(q, x) \sum_u p(u \mid x, q) log\left[\frac{p(u \mid x, q)}{p(u \mid q)}\right], \\
&= \sum_{q,x} p(q, x) \sum_u p(u \mid x) log\left[\frac{p(u \mid x)}{p(u \mid q)}\right], \\
&= E_{X,Q}\left[D_{KL}(p_{U|X} \mid\mid p_{U|Q})\right].
\end{aligned} \tag{3}
$$

Here we used the fact that $U$ is conditionally independent of $Q$ given $X$. We then observe that Eq. 3 induces the Kullback-Leibler divergence $D_{KL}$ as a natural transparency metric $\delta_u$ in Def.2.2.

Similarly, we can analyze $I(S; Q)$ to get,

$$
\begin{aligned}
I(S; Q) &= \sum_q p(q) \sum_s p(s \mid q) log\left[\frac{p(s \mid q)}{p(s)}\right], \\
&= E_{X,Q}\left[RD_{KL}(p_S \mid\mid p_{S|Q})\right].
\end{aligned} \tag{4}
$$

We can see from Eq.4 that the natural induced metric for measuring privacy $\delta_s$ in Def,2.1 is the reverse Kullback-Leibler divergence $RD_{KL}$.

We can thus rewrite our fundamental tradeoff equation as

$$E_{X,Q}[RD_{KL}(p_S \mid\mid p_{S|Q}) + D_{KL}(p_{U|X} \mid\mid p_{U|Q})] \geq I(U, S) - I(U, S \mid X). \tag{5}$$

We show next how this bound can be used to define a trainable loss metric, allowing the privatizer to select different points in the transparency-privacy trade-off space.

## 3.2 DEFINING A TRAINABLE LOSS METRIC

Assume that for any given stochastic transformation mapping $Q \sim Q(X)$, we have access to the posterior conditional probability distributions $P(S \mid Q)$, $P(U \mid Q)$, and $P(U \mid X)$. Assume we also have access to the prior distribution of $P(S)$. Inspired by the bounds from the previous section, the proposed privatizer loss is

$$min_Q E_{X,Q}[(1 - \alpha)D_{KL}(p_{U|X} \mid\mid p_{U|Q})^2 + \alpha RD_{KL}(p_S \mid\mid p_{S|Q})^2], \tag{6}$$

where $\alpha \in [0, 1]$ is a tradeoff constant. A low $\alpha$ value implies a high degree of transparency, while a high value of $\alpha$ implies a high degree of privacy. Using Eq.5 we have a lower bound on how private or transparent the privatizer can be for any given $\alpha$ value, as detailed next.

**Theorem 3.3.** For any $\alpha \in [0, 1]$, and stochastic mapping $Q : \mathcal{X} \rightarrow \mathcal{Q}$ the solution to Eq.6 guarantees the following bounds,

$$
\begin{aligned}
E_{X,Q}[D_{KL}(p_{U|X} \,||\, p_{U|Q})] &\geq \alpha[I(U, S) - I(U, S \mid X)], \\
E_{X,Q}[RD_{KL}(p_S \,||\, p_{S|Q})] &\geq (1 - \alpha)[I(U, S) - I(U, S \mid X)].
\end{aligned}
\tag{7}
$$

The proof is shown in Supplementary Material. To recap, we proposed a privatizer loss Eq.6 with a controllable trade-off parameter $\alpha$, and showed bounds on how transparent and private our data can be for any given value of $\alpha$. Next we show how to optimize this utility-privacy formulation.

# 4 A DATA-DRIVEN IMPLEMENTATION

Even if the joint distribution of $P(U, S, X)$ is not known, the privatizer can attempt to directly implement Eq.6 in a data-driven architecture to find the optimal $Q$. Assume the privatizer has access to a dataset $\{(x, s, u)\}$, where $s$ and $u$ are the ground truth secret and utility values of observation $x$. Under these conditions, the privatizer searches for a parametric mapping $q = Q_\theta(x, z)$, where $z$ is an independent random variable, and attempts to predict the best possible attack by learning $P_\eta(s \mid q)$, an estimator of $P(s \mid q)$. The privatizer also needs $P_\psi(u|q)$ and $P_\phi(u|x)$, estimators of $P(u \mid q)$ and $P(u \mid x)$ respectively, to measure how much information about the utility variable is censored with the proposed mapping. Under this setup $Q_\theta(x, z)$ is obtained by optimizing the following adversarial game:

$$
\begin{aligned}
\hat{\eta} &= \mathrm{argmin}_\eta E_{X,S,Z}\big[ - log(P_\eta(s|Q_{\hat{\theta}}(x, z)))\big], \\
\hat{\psi} &= \mathrm{argmin}_\psi E_{X,U,Z}\big[ - log(P_\psi(u|Q_{\hat{\theta}}(x, z)))\big], \\
\hat{\phi} &= \mathrm{argmin}_\phi E_{X,U}\big[ - log(P_\phi(u|x))\big], \\
\hat{\theta} &= \mathrm{argmin}_\theta (1 - \alpha)E_{X,U,Z}^2\big[D_{KL}(P_{\hat{\phi}}(u \mid x) \,||\, P_{\hat{\psi}}(u \mid Q_{\hat{\theta}}(x, z)))\big] + \\
&\quad + \alpha E_{X,S,Z}^2\big[RD_{KL}(P(s) \,||\, P_{\hat{\eta}}(s \mid Q_{\hat{\theta}}(x, z)))\big].
\end{aligned}
\tag{8}
$$

Here the first three terms are crossentropy loss terms to ensure our estimators $P_\eta(s|q)$, $P_\psi(u|q)$, and $P_\phi(u|x)$ are a good approximation to the true posterior distributions. The final loss term attempts to find the best possible sampling function $Q_\theta(x, z)$ such that $(1 - \alpha)I^2(U; X \mid Q) + \alpha I^2(S; Q)$ is minimized. Details on the algorithmic implementation are given in Section 7.3.1. Performance on simulated datasets is shown in Section 7.2.

## 4.1 PRIVACY UNDER FIXED UTILITY INFERENCE

The proposed framework naturally provides a means to achieve collaboration from the utility provider. In this scenario, the utility provider wishes to respect the user's desired privacy, but is unwilling to change their estimation algorithm $P_{\hat{\phi}}(u \mid x)$, and expects the privatizer to find a mapping that minimally affects its current performance.[2] This is a more challenging scenario, with worse tradeoff characteristics, in which $Q_\theta(x, z)$ is obtained by optimizing

$$
\begin{aligned}
\hat{\eta} &= \mathrm{argmin}_\eta E_{X,S,Z}\big[ - log(P_\eta(s|Q_{\hat{\theta}}(x, z)))\big], \\
\hat{\theta} &= \mathrm{argmin}_\theta (1 - \alpha)E_{X,U,Z}^2\big[D_{KL}(P_{\hat{\phi}}(u \mid x) \,||\, P_{\hat{\phi}}(u \mid Q_{\hat{\theta}}(x, z)))\big] + \\
&\quad + \alpha E_{X,S,Z}^2\big[RD_{KL}(P(s) \,||\, P_{\hat{\eta}}(s \mid Q_{\hat{\theta}}(x, z)))\big].
\end{aligned}
\tag{9}
$$

---

[2]Recall that the utility provider wants to use the same algorithm for sanitized and non-sanitized data, a unique aspect of the proposed framework and critical to accept its collaboration.

### 4.2 PRIVACY UNDER FIXED UTILITY AND SECRET INFERENCE

A final scenario addressed by the proposed framework arises when the utility provider is the sole agent to access the sanitized data, and it has estimation algorithms for both the utility and the privacy variable $P_{\hat{\phi}}(u \mid x)$, $P_{\hat{\eta}}(s \mid x)$, that it is unwilling to modify. The service provider wishes to reassure the users that they are unable to infer the secret attribute from the sanitized data, if and when the user decides so. Under these conditions, we optimize for

$$\hat{\theta} = \text{argmin}_\theta (1 - \alpha) E_{X,U,Z}^2 \big[ D_{KL}(P_{\hat{\phi}}(u \mid x) \mid\mid P_{\hat{\phi}}(u \mid Q_{\hat{\theta}}(x, z))) \big] + \\ + \alpha E_{X,S,Z}^2 \big[ R D_{KL}(P(s) \mid\mid P_{\hat{\eta}}(s \mid Q_{\hat{\theta}}(x, z))) \big]. \tag{10}$$

## 5 EXPERIMENTS AND RESULTS

The following examples are based on the framework presented in Figure 2. Here we have the three key agents mentioned before: (1) the utility algorithm that is used by the provider to estimate the information of interest. This algorithm can take the raw data ($X$) or the mapped data ($Q(X)$) and be able to infer the utility; (2) the secret algorithm that is able to operate on the mapped data to infer the secret; (3) the privatizer that learns a space preserving mapping $Q$ that allows the provider to learn the utility but prevents the secret algorithm to infer the secret. The utility algorithm is trained to perform well on raw data, the secret algorithm is adversarially trained to infer the secret variable after sanitization. In the next examples we show how the proposed framework performs under different scenarios, the privatizer architecture is kept unchanged across all experiments to show that the same architecture can achieve very different objectives using the proposed framework, the detailed architectures are shown in Section 7.3.2. Extra experiments under known conditions are shown in 7.2.

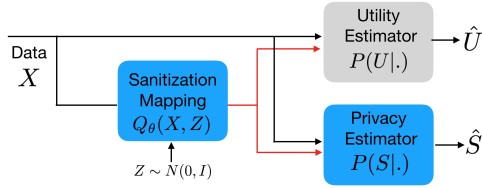

Figure 2: Three components of the collaborative privacy framework. Raw data can be directly fed into the secret and utility inferring algorithm. Since the privatization mapping is space preserving, the privatized data can also be directly fed to both tasks without any need for further adaptations.

### 5.1 SUBJECT WITHIN SUBJECT

We begin by analyzing the *subject-within-subject* problem. Imagine a subset of users wish to unlock their phone using facial identification, while others opt out of the feature; we wish the face identification service to work only on the consenting subset of users. We additionally assume that the utility provider wishes to comply with the user's wishes, so we can apply the framework described in Section 4.2. Note that in this problem, the utility and secrecy variables are mutually exclusive.

We solve this problem by training a space-preserving stochastic mapping $Q$ on facial image data $X$, where the utility and secret variable $U$ and $S$ are categorical variables over consenting and non-consenting users respectively. We test this over the FaceScrub dataset Kemelmacher-Shlizerman et al. (2016), using VGGFace2 Cao et al. (2017) as the utility and secrecy inferring algorithm. The stochastic mapping was implemented using a stochastic adaptation of the UNET Ronneberger et al. (2015), architecture details are provided in Section 7.3.2.

Table 1 shows the top-5 categorical accuracy of the utility network over the sanitized data at various $\alpha$ points in the privacy-utility trade-off. Figure 3 show some representantive images on how images are sanitized. It also shows that the sanitization function is able to preserve information about the utility variable while effectively censoring the secret variable, even for unobserved images. A phone equipped with this filter at the sensor level would be effectively incapable of collecting information on nonconsenting users.

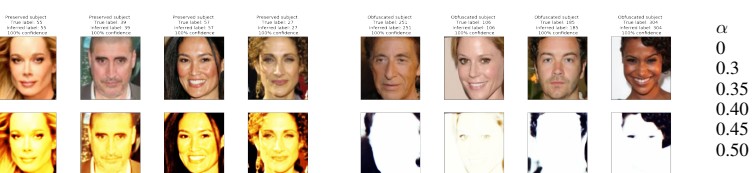

| | | Top-5 Accuracy | |
|---|---|---|---|
| $\alpha$ | CU | OPU | UPU |
| 0 | 99.8% | 99.8% | 99.8% |
| 0.3 | 94.7% | 86.1% | 87.5% |
| 0.35 | 93.5% | 15.3% | 33.6% |
| 0.40 | 93.0% | 14.9% | 31.8% |
| 0.45 | 89.4% | 12.6% | 30.2% |
| 0.50 | 70.7% | 10.9% | 27.0% |

(a) Filtered images of consenting users (CU)

(b) Filtered images of private users (PU)

Table 1: Subject detection on users

Figure 3: Left and center figures show images of consenting and nonconsenting (private) users respectively, along with their sanitized counterparts. The identity of consenting users is still easily verified, while the identity of nonconsenting users is effectively censored. Table on the right shows Top-5 accuracy performance of the subject detector after sanitization across several sanitation levels $\alpha$. Performance is shown across 3 subsets, consenting users (CU) are users that decided to be detected by the utility algorithm, observed private users (OPU) are those that explicitly decided to protect their privacy, while unobserved private users (UPU) are users that decided to protect their privacy but where not available during training. Consenting users are still recognized by the system, while nonconsenting users are not. For example, for $\alpha = 0.4$, we significantly block OPU and UPU while preserving CU

## 5.2 Obfuscating Emotion While Preserving Gender

Here we continue to work on facial image data $X$, where utility variable $U$ is gender recognition, and the secret variable $S$ is emotion (smiling/non-smiling). In this scenario, variables $U$ and $S$ are independent. We implement this over the CelebA dataset Liu et al. (2015), using Xception networks Chollet (2017) as our utility and privacy estimators. Table.2 shows the distribution of the utility and secrecy estimators over the sanitized data. Figure 4 shows example sanitized images. It is visually possible to identify the gender of the subject but not their emotion. Most importantly, the existing gender detection algorithm still performs correctly over the sanitized images.

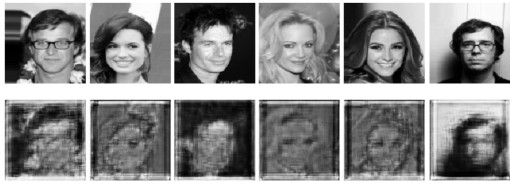

| Inference | Data | Male | Female |
|---|---|---|---|
| Gender | Raw | 94.2% | 94.5% |
| | Sanit. | 91.2% | 96.1% |
| Emotion | Raw | 93.3% | 92.0% |
| | Sanit. | 54.0% | 60.0% |

Table 2: Gender and emotion detection on users on raw and sanitized data.

Figure 4: Images before and after sanitization for Gender (utility) vs Emotion (privacy).

## 5.3 Subject preservation and gender obfuscation

In this setup, we want to find a mapping $Q$ that hides the gender attribute but allows subject verification. The mapping $Q$ should prevent a standard gender detection algorithm from performing its task, while allowing a standard subject detector algorithm to still perform subject verification. This is the only experiment in this section where the secret inference algorithm is fixed.

To realize this constrained scenario, we used a WideResNet-based Zagoruyko & Komodakis (2016) implementation of DEXNet Rothe et al. (2018) as our gender detector adversary, and a ResNet-50 implementation of VGGface2 Cao et al. (2017) as a subject identifier. The stochastic mapping was implemented using two differing architectures, the first is a stochastic adaptation of the UNET Ronneberger et al. (2015), and the second a concatenation of a FaderNet Lample et al. (2017) and a UNET Ronneberger et al. (2015) as shown in Bertran et al. (2018).

The mapping that incorporates a pretrained FaderNet was chosen as the baseline for the stochastic mapping function since this network is already trained to defeat a gender discriminator in its *encoding space*. This proves a suitable baseline comparison and starting point for a mapping function that needs to fool a gender discriminator *in image space* while simultaneously preserving subject verification performance. We show the performance of using only the pretrained gender FaderNet

and demonstrate how we can improve its performance by training a posterior processing mapping (UNET) using the loss proposed in Eq.10.

We tested this framework on the FaceScrub dataset Ng & Winkler (2014). Figure 5 shows how the output probabilities of the gender classification model approach the prior distribution of the dataset as $\alpha$ increases. We see that sanitized images produce output gender probabilities close to the dataset prior even for relatively low $\alpha$ values. Last column of figure 5 shows how the top-5 categorical accuracy of the subject verification task varies across different $\alpha$ values. These results suggest that under these conditions we can achieve almost perfect privacy while maintaining reasonable utility performance.

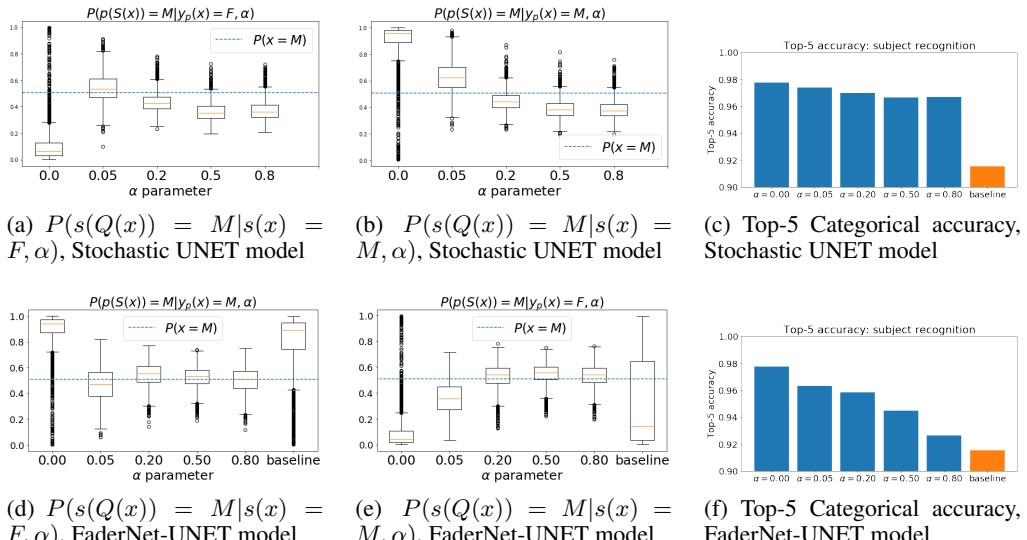

(a) $P(s(Q(x)) = M|s(x) = F, \alpha)$, Stochastic UNET model

(b) $P(s(Q(x)) = M|s(x) = M, \alpha)$, Stochastic UNET model

(c) Top-5 Categorical accuracy, Stochastic UNET model

(d) $P(s(Q(x)) = M|s(x) = F, \alpha)$, FaderNet-UNET model

(e) $P(s(Q(x)) = M|s(x) = M, \alpha)$, FaderNet-UNET model

(f) Top-5 Categorical accuracy, FaderNet-UNET model

Figure 5: First two columns show gender probabilities on the sanitized data $Q(X)$ as a function of $\alpha$, results are split according to real gender ($M$:Male, $F$:Female). Whisker plots show median and interquartile ranges, with outliers shown as circles. Pretrained FaderNet is shown as a baseline for comparison. Third column shows the top-5 categorical accuracy of the subject recognition task as a function of $\alpha$. Each row correspond to the Stochastic UNET and FaderNet-UNET models. Note that the concatenation the UNET with the FaderNet is able to improve performance on the Top-5 categorical accuracy metric when compared to the baseline model.

## 6  CONCLUDING REMARKS

Inspired by information-theory bounds on the privacy-utility trade-off, we introduced a new paradigm where users and entities collaborate to achieve both utility and privacy per a user's specific requirements. One salient feature of this paradigm is that it can be completely transparent – involving only the use of a simple user-specific privacy filter applied to user data – in the sense that it requires otherwise no modifications to the system infrastructure, including the service provider algorithmic capability, in order to achieve both utility and privacy.

Representative architectures and results suggest that a collaborative user-controlled privacy approach can be achieved. While the results here presented clearly show the potential of this approach, much has yet to be done, of particular note is extending this approach to continuous utility and privacy variables. While the underlying framework still holds, reliably measuring information between continuous variables is a more challenging task to perform and optimize for.

The proposed framework provides privacy metrics and bounds in expectation; we are currently investigating how the privacy tails concentrate as data is acquired and if there is a need to use information theory metrics with worst-case scenario guarantees. Modifying the information theory metrics to match some of the theoretical results in (local) differential privacy is also the subject of future research.

Privacy is closely related to fairness, transparency, and explainability, both in goals and in some of the underlying mathematics. A unified theory of these topics will be a great contribution to the ML community.

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

## 7 SUPPLEMENTARY MATERIAL

### 7.1 PROOFS

*Proof.* Lemma 1

Consider the equality

$$
\begin{aligned}
I(U;S) - I(U;S|X) = {} & I(S;Q(X)) - I(S;Q(X)|U) \\
& + I(U;X|Q(X)) - I(U;X|Q(X),S).
\end{aligned}
\tag{11}
$$

We know that

$$
0 \leq I(S;Q(X)|U) \leq I(S;Q(X)), \quad 0 \leq I(U;X|Q(X),S) \leq I(U;X|Q(X)),
\tag{12}
$$

so we can guarantee

$$
I(U;S) - I(U;S|X) \leq I(S;Q(X)) + I(U;X|Q(X)).
\tag{13}
$$

$\square$

*Proof.* Theorem 3.3

Consider

$$
\begin{aligned}
A_Q &= E_{X,Q}[D_{KL}(p_{U|X} \,||\, p_{U|Q})], \\
B_Q &= E_{X,Q}[D_{KL}(p_{U|X} \,||\, p_{U|Q})], \\
K &= I(U,S) - I(U,S \mid X),
\end{aligned}
\tag{14}
$$

and $\alpha \in [0,1]$.

Minimizing Eq.6 respecting Eq.5 and Eq.12 is equivalent to solving:

$$
\begin{aligned}
& min_Q\{(1-\alpha)A_Q^2 + \alpha B_Q^2\}, \\
& s.t. A_Q, B_Q \geq 0, \\
& \quad A_Q + B_Q \geq K, \\
& \quad \alpha \in [0,1],
\end{aligned}
\tag{15}
$$

Consider the following relaxation of Eq.15

$$
\begin{aligned}
& min_{A,B}\{(1-\alpha)A^2 + \alpha B^2\}, \\
& s.t. A, B \geq 0, \\
& \quad A + B \geq K, \\
& \quad \alpha \in [0,1],
\end{aligned}
\tag{16}
$$

where $A$ and $B$ are positive real values. Eq.16 is a relaxation of Eq.15 because the space of possible tuples $(A_Q, B_Q)$ is included in the space of possible values of $R^{2+}$

Suppose $Q^*$ is the solution to Eq.15, with corresponding values $(A_{Q^*}, B_{Q^*})$, and suppose $(A^*, B^*)$ is the solution to Eq. 16. We know

$$
(1-\alpha)A^{*2} + \alpha B^{*2} \leq (1-\alpha)A_{Q^*}^2 + \alpha B_{Q^*}^2,
\tag{17}
$$

Assume $A^* > A_{Q^*}$, it follows that $(1-\alpha)A_{Q^*} + \alpha B^{*2} \leq (1-\alpha)A^{*2} + \alpha B^{*2}$.

However, $A_{Q^*} > 0$ and $A_{Q^*} + B^* \leq A^* + B^* \leq K$. Therefore, $(A_{Q^*}, B^*)$ is a valid solution to Eq.16, and is smaller than the lower bound $(A^*, B^*)$.

This contradiction arises from assuming $A^* > A_{Q^*}$, we thus conclude that

$$A^* \leq A_{Q^*}.$$

Similarly for $B^*$ and $B_{Q^*}$ we get

$$B^* \leq B_{Q^*}.$$

Additionally, Eq.16 is easily solvable and has solutions

$$A^* = \alpha K; \qquad B^* = (1 - \alpha)K \tag{18}$$

Consequently, we proved

$$E_{X,Q}[D_{KL}(p_{U|X} \parallel p_{U|Q})] \geq \alpha[I(U, S) - I(U, S \mid X)],$$
$$E_{X,Q}[RD_{KL}(p_S \parallel p_{S|Q})] \geq (1 - \alpha)[I(U, S) - I(U, S \mid X)]. \tag{19}$$

$\square$

## 7.2 EXPERIMENTS ON SIMULATED DATA

The following experiments attempt to show how close to the theoretical bound shown in Eq. 5 we can get by following Algorithm 1 under known conditions.

Consider the following scenario: Utility variable $U$ and secret variable $S$ are two Bernoulli variables with the following joint distribution:

$$P(u, s) = \begin{cases} k\rho\beta & \text{if } (u, s) = (1, 1) \\ (1 - k\rho)\beta & \text{if } (u, s) = (0, 1) \\ (1 - k\beta)\rho & \text{if } (u, s) = (1, 0) \\ 1 - \beta - \rho + k\rho\beta & \text{if } (u, s) = (1, 0) \end{cases} \tag{20}$$

where the marginal probabilities are $P(U = 1) = \rho$, $P(S = 1) = \beta$, and $k$ is a parameter that controls the dependence between and $S$, $k \in [0, min\{\rho^{-1}, \beta^{-1}\}]$.

For these experiments, we make both marginals equal to 0.5 ($\rho = \beta = 0.5$). Note that when $k = 1$, $U$ and $S$ are independent ($I(U; S) = 0$) and when $k = 0$ or $k = 2$ they reach maximum mutual information ($I(U; S) = H(U) = H(S) = H_b(0.5) = ln(1)$ nats).

Our observations $X$ will be taken in the extreme case, where $X$ contains almost perfect information about the values of $U$ and $S$. We do this by assuming that $X \in R^2$ is a Gaussian Mixture Model $X$ with the following conditional distribution:

$$X|U, S \sim N(\begin{bmatrix} U \\ S \end{bmatrix}, \sigma^2 I_{2 \times 2}) \tag{21}$$

We choose a low $\sigma = 0.05$; this makes it so that every pair $(u, s)$ is mapped to almost entirely disjoint regions, therefore knowing $X$ gives nearly perfect information about $(u, s)$ ($I(U; X) \simeq H(U), I(S; X) \simeq H(S), I(U; S|X) \simeq 0$). For added simplicity, the privacy filter is linear:

$$Q_\theta(x, z) = \theta \times x + z$$
$$Z \sim N(\begin{bmatrix} 0 \\ 0 \end{bmatrix}, I_{2 \times 2}) \tag{22}$$

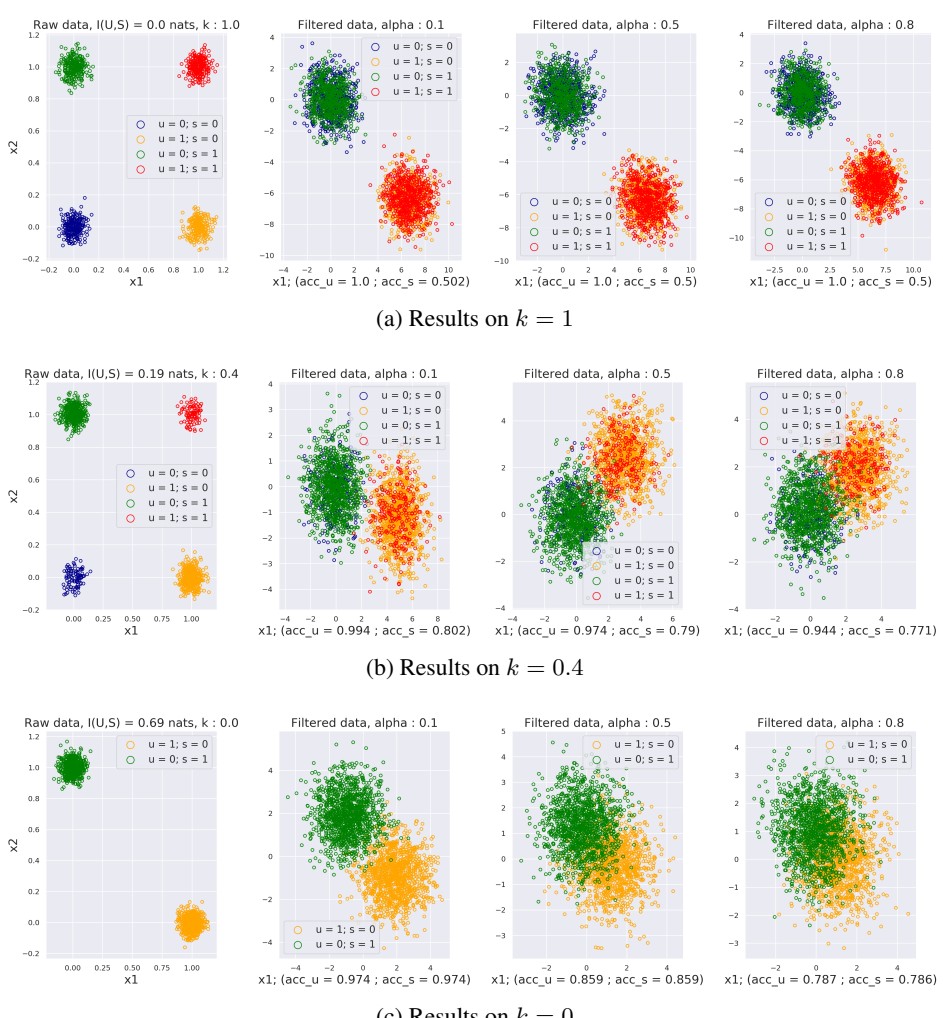

Figure 6: Left column shows the baseline distributions across varying codependence levels $I(U, S) = \{0, 0.19, 0.69\}$[nats] ($k = \{1, 0.4, 0\}$). Remaining columns show the linearly sanitized data for varying levels of tradeoff $\alpha$, from left to right, $\alpha = \{0.1, 0.5, 0.8\}$. As $\alpha$ increases, the amount of utility the filter allows gets gradually smaller.

Figure 6 shows how the raw and sanitized data are distributed for varying levels of codependence $k$ and tradeoff $\alpha$ for linear sanitization functions. Figure 7 shows that privacy filters optimized using Algorithm 1 learn effective privacy-preserving mappings close to the theoretical bounds, even for a simple filtering architecture. They do so without any explicit modelling of the underlying data-generating distributions, and we can achieve different tradeoff points by simply modifing the parameter $\alpha$. Note that when variables $U$ and $S$ are perfectly independent or codependent, the linear filter is perfectly able to reach any point in the optimal bound. For intermediate cases, the linear filter was capable of reaching the bound in the region where no utility is compromised, but was not capable of following the optimal tradeoff line for higher levels of privacy.

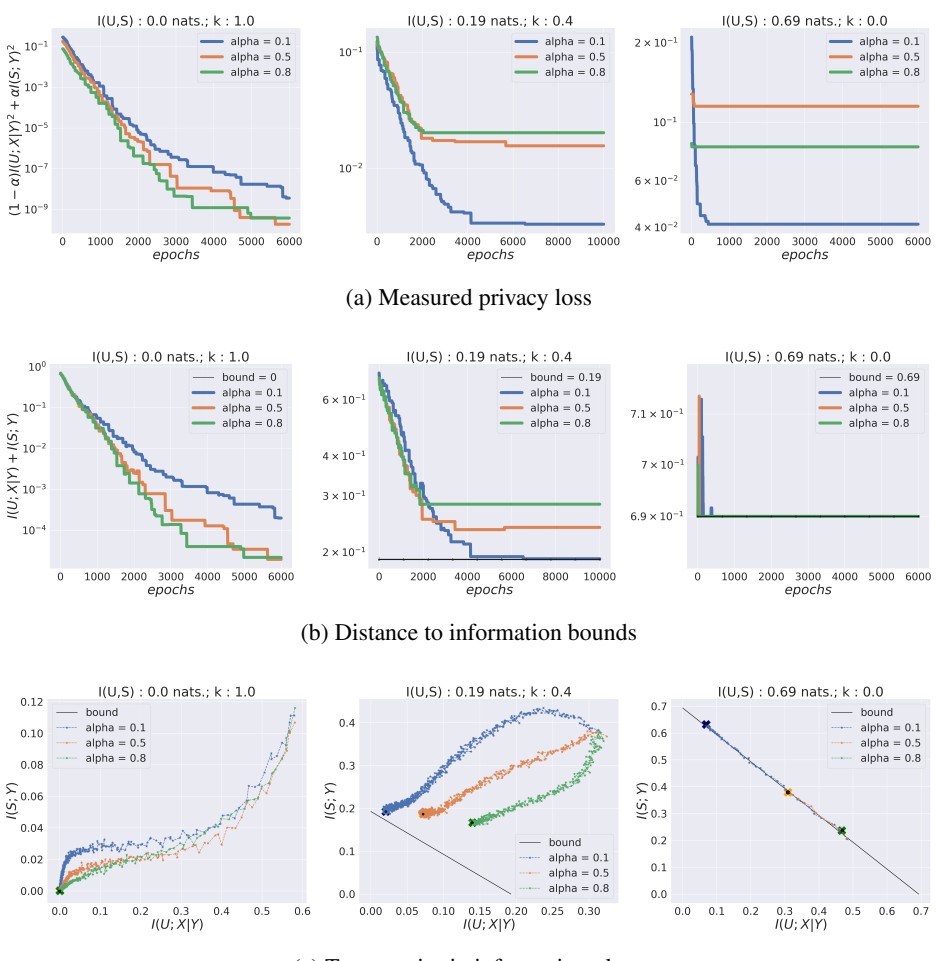

(a) Measured privacy loss

(b) Distance to information bounds

(c) Trayectories in information plane

Figure 7: Top row shows the best privacy-utility loss so far on the validation set for different levels of codependence $I(U; S)$ and trade-off parameter $\alpha$. Middle row shows how the sum of the estimated informations approximate the information bound for the best privacy-utility loss so far on the validation set. Finally, bottom row illustrates the trajectory in information space of the privacy-utility filters as they are being trained.

## 7.3 IMPLEMENTATION DETAILS

Here, we explicitly elaborate on how we optimize the data-driven loss functions shown in equations 8, 9, and 10 using an adversarial approach. We first detail the exact adversarial training setup that was used to perform the experiments in Section 5, and then provide the concrete network architectures used for all shown results.

### 7.3.1 ALGORITHMIC IMPLEMENTATION

Minimizing the objective functions in equations 8, 9, and 10, is a challenging problem in general. By focusing our attention on Eq. 8, we see that each of the four loss terms have a distinct purpose:

$$\hat{\eta} = argmin_\eta E_{X,S,Z}\big[-log(P_\eta(s|Q_{\hat{\theta}}(x,z))\big],$$
$$\hat{\psi} = argmin_\psi E_{X,U,Z}\big[-log(P_\psi(u|Q_{\hat{\theta}}(x,z))\big],$$
$$\hat{\phi} = argmin_\phi E_{X,U}\big[-log(P_\phi(u|x)\big], \tag{23}$$
$$\hat{\theta} = argmin_\theta(1-\alpha)E^2_{X,U,Z}\big[D_{KL}(P_{\hat{\phi}}(u\mid x)\mid\mid P_{\hat{\psi}}(u\mid Q_{\hat{\theta}}(x,z)))\big]+$$
$$+\alpha E^2_{X,S,Z}\big[RD_{KL}(P(s)\mid\mid P_{\hat{\eta}}(s\mid Q_{\hat{\theta}}(x,z)))\big].$$

The first three loss terms minimize a crossentropy objective for functions $P_\eta(s\mid q)$, $P_\psi(u\mid q)$, and $P_\phi(u\mid x)$; this ensures that these functions are good estimators of the unknown true distributions of $P(s\mid q)$, $P(u\mid q)$, and $P(u\mid x)$, where samples $q$ are drawn from the learned sanitization mapping $q = Q_\theta(x,z)$. The final loss term attempts to find the best possible sampling function $Q_\theta(x,z)$ such that $(1-\alpha)I^2(U;X\mid Q) + \alpha I^2(S;Q)$ is minimized.

We can approximately solve this problem by applying iterative Stochastic Gradient Descent to each of the individual loss terms with respect to their relevant parameters, this is similar to the procedure used to train Generative Adversarial Networks.

The algorithm we used to solve Eq. 8 is shown in Algorithm 1, similarly, algorithms to solve Eq. 9 and Eq. 10 are shown in Algorithm 2 and Algorithm 3 respectively.

---

**Algorithm 1** Sanitization algorithm

$(\theta, \eta, \phi, \psi) \leftarrow$ initialize network parameters

1: **repeat**
2:     $(x^{(1)}, u^{(1)}, s^{(1)}), ...(x^{(b)}, u^{(b)}, s^{(b)}) \sim P_{X,U,S}$     $\triangleright$ Draw $b$ samples from joint distribution
3:     $z^{(1)}, ...z^{(b)} \sim P_Z$     $\triangleright$ Draw $b$ samples from sampling distribution
4:     $\Phi(\phi) = \frac{1}{b}\sum_{i=1}^b -logP_\phi(u^i\mid x^i)$     $\triangleright$ Evaluate crossentropy loss on raw utility inference
5:     $\phi \leftarrow \phi - lr\nabla_\phi\Phi(\phi)$     $\triangleright$ Stochastic gradient descent step on $P_\phi(u\mid x)$
6:     $\Psi(\psi) = \frac{1}{b}\sum_{i=1}^b -logP_\psi(u^i\mid Q_\theta(x^i,z^i))$     $\triangleright$ Evaluate crossentropy loss on filtered utility inference
7:     $\psi \leftarrow \psi - lr\nabla_\psi\Psi(\psi)$     $\triangleright$ Stochastic gradient descent step on $P_\psi(u\mid q)$
8:     $H(\eta) = \frac{1}{b}\sum_{i=1}^b -logP_\eta(s^i\mid Q_\theta(x^i,z^i))$     $\triangleright$ Evaluate crossentropy loss on secret inference
9:     $\eta \leftarrow \eta - lr\nabla_\eta H(\eta)$     $\triangleright$ Stochastic gradient descent step on $P_\eta(s\mid q)$
10:    $\Theta(\theta) = (1-\alpha)\big[\frac{1}{b}\sum_{i=1}^b log\frac{P_\phi(u^i|x^i)}{P_\psi(u^i|Q_\theta(x^i,z^i))}\big]^2 + \alpha\big[\frac{1}{b}\sum_{i=1}^b log\frac{P_\eta(s^i|Q_\theta(x^i,z^i)))}{P(s^i)}\big]^2$     $\triangleright$ Evaluate sanitation loss
11:    $\theta \leftarrow \theta - lr\nabla_\theta\Theta(\theta)$     $\triangleright$ Stochastic gradient descent step on $Q_\theta(x,z)$
12: **until** convergence

---

### 7.3.2 NETWORK ARCHITECTURES

We now describe the exact architecture used to implement the privacy filter on all experiments shown in Section 5. Figure 8 shows the network diagram.

The architecture presented in Figure 8 is fully convolutional, so the same network definition could be used across all three experiments by varying the input layer. To speed up convergence to a good filtering solution, filters were initially trained to copy the image (under RMSE loss), and optionally infer some meaningful attribute from the input (in *subject-within-subject*, this attribute was a simple class label on whether the subject wished their privacy preserved). We stress that this was only done for initialization, final training of the network was done exactly as described in Algorithm 1.

---

**Algorithm 2** Sanitization algorithm; fixed utility

---

$(\theta, \eta) \leftarrow$ initialize network parameters

$\phi \leftarrow \hat{\phi}$ Utility inference algorithm is fixed and known

1: **repeat**
2:     $(x^{(1)}, u^{(1)}, s^{(1)}), ...(x^{(b)}, u^{(b)}, s^{(b)}) \sim P_{X,U,S}$       ▷ Draw $b$ samples from joint distribution
3:     $z^{(1)}, ...z^{(b)} \sim P_Z$                             ▷ Draw $b$ samples from sampling distribution
4:     $H(\eta) = \frac{1}{b} \sum_{i=1}^{b} -log P_\eta(s^i | Q_\theta(x^i, z^i))$    ▷ Evaluate crossentropy loss on secret inference
5:     $\eta \leftarrow \eta - lr \nabla_\eta H(\eta)$                         ▷ Stochastic gradient descent step on $P_\eta(s \mid q)$
6:     $\Theta(\theta) = (1 - \alpha) \left[ \frac{1}{b} \sum_{i=1}^{b} log \frac{P_\phi(u^i | x^i)}{P_\phi(u^i | Q_\theta(x^i, z^i))} \right]^2 + \alpha \left[ \frac{1}{b} \sum_{i=1}^{b} log \frac{P_\eta(s^i | Q_\theta(x^i, z^i)))}{P(s^i)} \right]^2$       ▷
    Evaluate sanitation loss
7:     $\theta \leftarrow \theta - lr \nabla_\theta \Theta(\theta)$                         ▷ Stochastic gradient descent step on $Q_\theta(x, z)$
8: **until** convergence

---

**Algorithm 3** Sanitization algorithm; fixed utility and secret

---

$\theta \leftarrow$ initialize network parameters

$\phi \leftarrow \hat{\phi}$ Utility inference algorithm is fixed and known

$\eta \leftarrow \hat{\eta}$ Secret inference algorithm is fixed and known

1: **repeat**
2:     $(x^{(1)}, u^{(1)}, s^{(1)}), ...(x^{(b)}, u^{(b)}, s^{(b)}) \sim P_{X,U,S}$       ▷ Draw $b$ samples from joint distribution
3:     $z^{(1)}, ...z^{(b)} \sim P_Z$                             ▷ Draw $b$ samples from sampling distribution
4:     $\Theta(\theta) = (1 - \alpha) \left[ \frac{1}{b} \sum_{i=1}^{b} log \frac{P_\phi(u^i | x^i)}{P_\phi(u^i | Q_\theta(x^i, z^i))} \right]^2 + \alpha \left[ \frac{1}{b} \sum_{i=1}^{b} log \frac{P_\eta(s^i | Q_\theta(x^i, z^i)))}{P(s^i)} \right]^2$       ▷
    Evaluate sanitation loss
5:     $\theta \leftarrow \theta - lr \nabla_\theta \Theta(\theta)$                         ▷ Stochastic gradient descent step on $Q_\theta(x, z)$
6: **until** convergence

---

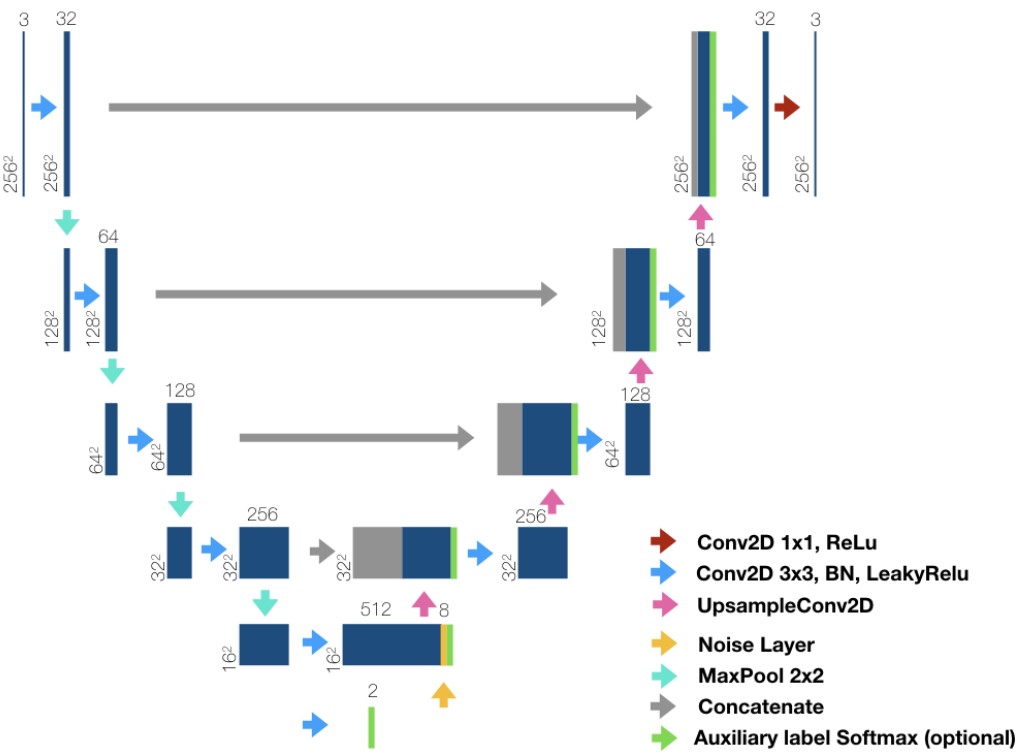

Figure 8: Architecture of privacy filter, based on UNET. There is a single noise layer (shown in yellow) where standard Gaussian noise is injected into the network to make the resulting filtered image stochastic in nature. The other notable component is the auxiliary label softmax, used for the *subject-within-subject* experiment. This extra layer was trained only to initialize the network, but was not preserved during the final training stage. Input image sizes are shown for the *subject-within-subject* experiment.

The architecture of the networks used to infer the utility and secret attribute in the emotion vs. gender experiment are identical, and are shown in Figure 9.

Networks used for the experiments in Section 7.2 are shown in Figure 10.

All other networks used in the results section are implemented as described in their respective papers.

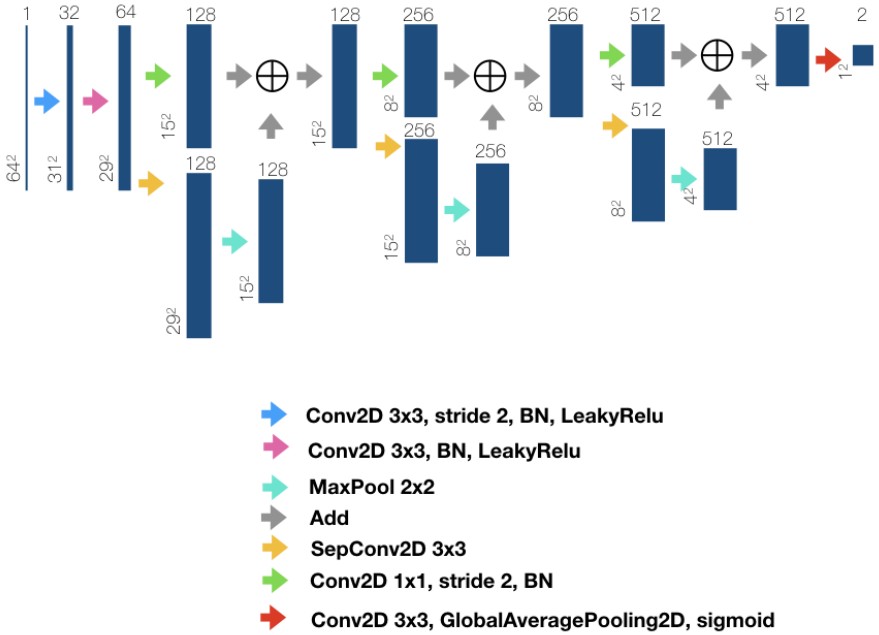

Figure 9: Architecture of utility and secret inference networks used in the emotion vs gender experiment. These architectures closely follow the one proposed in Chollet (2017)

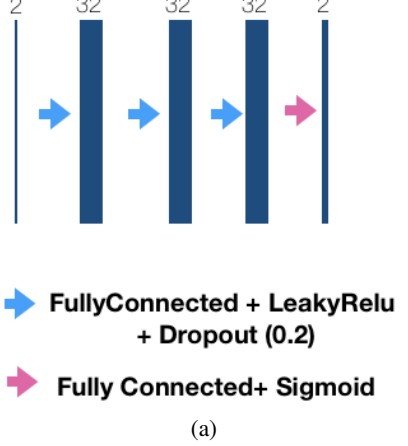

(a)

Figure 10: Architecture used for utility and secret inference in the experiments described in Section 7.2

