# OpenReview forum: "Learning data-derived privacy preserving representations from information metrics"
_ICLR.cc/2019/Conference_

### Official Review · AnonReviewer2 · 2018-11-02
**Interesting ideas, some major practical limitations**

**Rating:** 6
**Confidence:** 3

**Review:**

[Second update] I'd like to thank the authors for their detailed response. The authors have made changes that I believe improve the overall quality of the submission. I now lean towards accepting the paper, and have increased my rating from a 5 to a 6.

Most notably: (i) they clarified that their secret-detection model was retrained on sanitized data in their experiments, (ii) they added details about their experimental setup and the algorithms used for their experimental evaluation, and (iii) they added experiments to the appendix of the submission that evaluated their framework on synthetic data. I do, however, still have some concerns about how well the privacy guarantees of the proposed algorithm would hold up in practice against a motivated adversary (since formal privacy guarantees appear to be relatively weak right now).

As a minor comment, there may be a typo in Equation 20 of Section 7.2: the case (u, s) = (1, 0) is handled twice, whereas the case (u, s) = (0, 0) is never handled at all.

[First update] I find the authors' problem statement appealing, but share concerns with Reviewer 1 about the privacy guarantees offered by the proposed method, and with Reviewer 3 about need to clarify the experimental evaluation. No author response was provided; I've left my score for the paper unchanged. (Note: this update was posted a few days before the end of the rebuttal period; the submission was subsequently updated.)

[Summary] The authors consider a problem related to de-identification, where the goal is to perturb a dataset X in a way that makes it possible to infer some useful information U about each example in the dataset while obscuring some sensitive information S. For example, the authors consider the problem of perturbing pictures of people's faces to obfuscate the subjects' emotions while making it possible to infer their genders. The concrete approach explored in the paper's experimental evaluation ensures that an existing model trained model on the original dataset will continue to work when applied to the perturbed data.

On the theory side, the authors derive information-theoretic lower bounds on the extent to which one can disclose useful information about a dataset without leaking sensitive information, and propose concrete minimization problems that can perturb the data to trade off between the two objectives. On the practical side, the authors evaluate the minimization setup on three different problems.

[Key Comments] I'm of two minds about this paper. On the whole, I found the problem statement compelling. However, I had serious reservations about the implementation. First: I had trouble understanding the experimental setup based on the limited information provided in Section 5, and the results seem difficult to reproduce from the information in the paper. Second and more seriously: the security guarantees provided in practice seem very weak. At the very least, the authors should check whether their perturbations are robust against an adversary who retrains their model from scratch on perturbed data. This experiment would significantly strengthen the submission, but would still leave open the possibility that a clever adversary could extract more sensitive information than expected from the perturbed data.

[Details]
[Pro #1] The idea of perturbing an input in order to optimize bounds on how much "useful" versus "secret" information is disclosed by the output seems intuitively appealing. In that context, the theory from Sections 2 and 3 seems well-motivated. Section 3.2 ("defining a trainable loss metric") is especially well-motivated. It provides a concrete objective function which, when minimized, can obfuscate data in a way that trades off between utility and secrecy.

[Pro #2] The idea of perturbing a dataset in a way that allows existing useful algorithms to continue working without modifications seems like an interesting and novel contribution. I found the following excerpt from the introduction especially compelling: "it is important to design collaborative systems where each user shares a sanitized version of their data with the service provider in such a way that user-defined non-sensitive tasks can be performed but user-defined sensitive ones cannot, without the service provider requiring to change any data processing pipeline otherwise."

[Pro #3] The paper combines theoretical results with empirical case studies on three different problems. Based on visual inspection, the outputs of the perturbation heuristics shown in Section 5 / Figure 3 and Figure 4 seem reasonable.

[Con #1] Few details are provided about the experimental setup used in Section 5, and it was difficult for me to understand how the theoretical results in Section 4 were actually being applied. There's typically a lot of work that goes into turning a theoretical objective function (e.g., Equation 10 in Section 4.2) into a practical experimental setup. This could be a major contribution of the paper. But right now, I feel like there aren't enough details about the implementation for me to reproduce the experiments.

[Con #2] I had trouble understanding the motivation for the Subject within Subject case study in Section 5.1. The authors describe the problem as follows: "Imagine a subset of users wish to unlock their phone using facial identification, while others opt instead to verify their right to access the phone using other methods; in this setting, we would wish the face identification service to work only on the consenting subset of users, but to respect the privacy of the remaining users." The proposed solution (Figure 3) applies minor perturbations to the pictures of consenting subjects while editing the photos of the non-consenting users to leave only their silhouettes. A simple baseline would be to remove the photos of the non-consenting users from the dataset entirely. The case study would greatly benefit from a discussion of why the baseline is insufficient. It's also perfectly reasonable to say that the section is meant as a way to check whether the objective function from Section 4 can lead to reasonable behavior in practice, but if so, the intent should be clarified.

[Con #3] As far as I can tell, the practical experiments in Section 5 assume that the party who perturbs the dataset knows exactly what algorithm an attacker will use to infer secret information. They also seem to assume that the attacker cannot switch to a different algorithm -- or even retrain an existing machine-learned model -- to try and counter the perturbation heuristics. From the beginning of Section 5: "Initially, we assume that the secret algorithm is not specifically tailored to attack the proposed privatization, but instead is a robust commonly used algorithm trained on raw data to infer the secret." Unless I missed something, it seems like this assumption is used throughout the experimental evaluation.

To the authors' credit, the submission states this assumption explicitly in Section 5. From a security perspective, however, this seems like a dangerous assumption to rely on, as it leaves "sanitized" data vulnerable to attacks. For example, an attacker with knowledge of the perturbation algorithm can retrain the model they use to extract sensitive information, using perturbed images in place of the original images in their training dataset.

My main practical concern is that the security guarantees provided by the submission seem fragile. It may be much easier to build a perturbation algorithm that is resistant to a single (known) attack than to remove the sensitive information from the dataset entirely. Right now, the empirical results in the submission seem to focus on the former.

---

> ### Author Response · Authors · 2018-11-23
> **Response to Reviewer**
>
> We are very excited with the very positive and enthusiastic support of all the reviewers, and their outstanding feedback. We thank the reviewers for the very constructive comments. We have taken all the reviewers' points into consideration and have modified the paper accordingly, see blue text in revised document for the major changes (all related to clarifications, following the ICLR policy). A detailed point-by-point answer is provided below. Overall, the paper was revised for clarity, adding additional intuition to the theory and applicability of the framework. We also clarified the purpose of some of the experiments, and added material in the Supplementary section to show additional examples on controlled situations (to further explain the bounds and their relevance), as well as extensive implementation details.
>
> #REVIEWER QUOTE:
> ‘[Key Comments] I'm of two minds about this paper. On the whole, I found the problem statement compelling. However, I had serious reservations about the implementation. First: I had trouble understanding the experimental setup based on the limited information provided in Section 5, and the results seem difficult to reproduce from the information in the paper. Second and more seriously: the security guarantees provided in practice seem very weak. At the very least, the authors should check whether their perturbations are robust against an adversary who retrains their model from scratch on perturbed data. This experiment would significantly strengthen the submission, but would still leave open the possibility that a clever adversary could extract more sensitive information than expected from the perturbed data.’
> #
>
> *RESPONSE:
> We appreciate your comments. First, we added a detailed implementation section to make explicitly clear how anyone can implement and reproduce the results shown in the paper, this is now shown in Section 7.3.
>
> As for the second point, the secret inference algorithm is indeed retrained from scratch on perturbed data, in all but one of the presented experiments, we apologize for not making this clearer from the submission. Paragraph 1 in Experiments and Results (page 6) was modified accordingly, since the original passage was confusing (the original comment was alluding to the initialization of the networks before starting the sanitization learning algorithm). Paragraph 1 in Page 7 was added to clarify this is the only experiment shown with fixed secret inference.
>
> Note that we perform very different experiments and utility/privacy cases with the same proposed framework.We expect this and the multiple clarifications and additions in the revised version (see next) addresses all these constructive comments.
>
> #REVIEWER QUOTE:
> ‘[Pro #1] The idea of perturbing an input …’
> ‘[Pro #2] The idea of perturbing a dataset …’
> ’[Pro #3] The paper combines theoretical results with …‘
> #
>
> *RESPONSE:
> We thank the reviewer for pointing out these pros, which we believe (in particular now that all the cons have been clarified/addressed) significantly outpaces the cons below.
>
> #REVIEWER QUOTE:
> ‘[Con #1] Few details are provided about the experimental setup used in Section 5, and it was difficult for me to understand how the theoretical results in Section 4 were actually being applied. There's typically a lot of work that goes into turning a theoretical objective function (e.g., Equation 10 in Section 4.2) into a practical experimental setup. This could be a major contribution of the paper. But right now, I feel like there aren't enough details about the implementation for me to reproduce the experiments.
> #
>
> *RESPONSE:
> We agree with the sentiment that the experimental implementation was not sufficiently explained, to that effect, we added  Section 7.3 in the supplementary material to show exactly how the loss is converted into an experimental setup. We apologize for not providing details in the original version. Code will also be released with the paper publication.

---

> > ### Author Response · Authors · 2018-11-23
> > **Response to Reviewer continued**
> >
> > #REVIEWER QUOTE:
> > [Con #2] I had trouble understanding the motivation for the Subject within Subject case study in Section 5.1. The authors describe the problem as follows: "Imagine a subset of users wish to unlock their phone using facial identification, while others opt instead to verify their right to access the phone using other methods; in this setting, we would wish the face identification service to work only on the consenting subset of users, but to respect the privacy of the remaining users." The proposed solution (Figure 3) applies minor perturbations to the pictures of consenting subjects while editing the photos of the non-consenting users to leave only their silhouettes. A simple baseline would be to remove the photos of the non-consenting users from the dataset entirely. The case study would greatly benefit from a discussion of why the baseline is insufficient. It's also perfectly reasonable to say that the section is meant as a way to check whether the objective function from Section 4 can lead to reasonable behavior in practice, but if so, the intent should be clarified.
> > #
> >
> > *RESPONSE:
> > We modified paragraph 1 of the Experiments and results section to better motivate the scenarios shown, and explain how the same framework and architecture can achieve very different but consistent results when faced with different privacy tasks.
> >
> > Paragraph 3 on page 7 was also modified to better motivate the subject-within-subject example in particular. The goal of this task is to essentially make the phone incapable of collecting data on non-consenting users after the privacy filter is deployed.
> >
> > #REVIEWER QUOTE:
> > [Con #3] As far as I can tell, the practical experiments in Section 5 assume that the party who perturbs the dataset knows exactly what algorithm an attacker will use to infer secret information. They also seem to assume that the attacker cannot switch to a different algorithm -- or even retrain an existing machine-learned model -- to try and counter the perturbation heuristics. From the beginning of Section 5: "Initially, we assume that the secret algorithm is not specifically tailored to attack the proposed privatization, but instead is a robust commonly used algorithm trained on raw data to infer the secret." Unless I missed something, it seems like this assumption is used throughout the experimental evaluation.
> > To the authors' credit, the submission states this assumption explicitly in Section 5. From a security perspective, however, this seems like a dangerous assumption to rely on, as it leaves "sanitized" data vulnerable to attacks. For example, an attacker with knowledge of the perturbation algorithm can retrain the model they use to extract sensitive information, using perturbed images in place of the original images in their training dataset.
> > My main practical concern is that the security guarantees provided by the submission seem fragile. It may be much easier to build a perturbation algorithm that is resistant to a single (known) attack than to remove the sensitive information from the dataset entirely. Right now, the empirical results in the submission seem to focus on the former.
> > #
> >
> > *RESPONSE:
> > We apologize for the confusion, all experiments except the subject vs gender experiment were done while adversarially training the secret, exactly for the reasons you stated above. The phrase you highlighted was modified in Paragraph 1 in Experiments and Results (page 6) since the original one was clearly confusing.  (the original comment was alluding to the initialization of the networks before starting the sanitization learning algorithm). Paragraph 1 in Page 7 was added to clarify this is the only experiment shown with fixed secret inference. We hope this clarifies this issue now.
> >
> > To conclude, we have addressed all the reviewer’s comments, in particular, the 3 mentioned Cons, and we hope he/she will now support accepting this paper now. While as with most papers there is still significant work to be done, the paper proposes a new important framework for privacy with new results and theory (the reviewer him/herself points out to the importance of this work). The reviewer clearly states he/she likes the ideas of the paper, and the cons mentioned (all very constructive, thanks) have all been carefully addressed in the revision.

---

> ### Author Response · Authors · 2018-12-07
> **Additional response**
>
> We would like to thank the reviewer for their comments on the updated manuscript.
>
> We addressed the typo in Equation 20.
>
> Regarding the comments on stronger privacy guarantees:
>
> Indeed we would like to get tighter and stronger guarantees, this is a starting point. We do feel that the type of privacy concerns that can be addressed in a way similar to the one presented in this paper can be of use in real-world scenarios. We will continue working in exactly that direction, that of providing formal guarantees and bounds on privacy provided by these types of approaches.

---

### Official Review · AnonReviewer1 · 2018-11-05
**Privacy preserving data representation**

**Rating:** 5
**Confidence:** 4

**Review:**

This paper studies the problem of representing data records with potentially sensitive information about individuals in a privacy-preserving fashion such that they can be later used for training learning models. Informally, it is expected from the transformed output of data record, one should be able to learn about a desired hidden variable, but should not be able to learn anything about a sensitive hidden variable. To that end, the paper proposes a KL divergence based privacy notion, and an algorithmic approach to learn a representation while balancing the utility privacy trade-off.

I am excited about the choice of the problem, but I have reservations about the treatment of privacy in the paper. First, KL divergence is a very weak (average case) notion privacy that can be easily broken. Second, the algorithm that is outlined in the paper gives an empirical way to compute the representation while balancing the utility-privacy trade-off (Eq. 6). However, there is no formal privacy guarantee for the algorithm. It is important to remember that unlike the utility, privacy is a worst-case notion and should formally hold in all occasions.

---

> ### Author Response · Authors · 2018-11-23
> **Response to reviewer**
>
> We are very excited about the very positive and enthusiastic support of all the reviewers, and their outstanding feedback. We thank the reviewers for the very constructive comments. We have taken all the reviewers' points into consideration and have modified the paper accordingly, see blue text in the revised document for the major changes (all related to clarifications, following the ICLR policy). A detailed point-by-point answer is provided below. Overall, the paper was revised for clarity, adding additional intuition to the theory and applicability of the framework. We also clarified the purpose of some of the experiments and added material in the Supplementary section to show additional examples on controlled situations (to further explain the bounds and their relevance), as well as extensive implementation details.
>
> # REVIEWER QUOTE:
> ‘This paper studies the problem of representing data records with potentially sensitive information about individuals in a privacy-preserving fashion such that they can be later used for training learning models. Informally, it is expected from the transformed output of data record, one should be able to learn about a desired hidden variable, but should not be able to learn anything about a sensitive hidden variable. To that end, the paper proposes a KL divergence based privacy notion, and an algorithmic approach to learn a representation while balancing the utility privacy trade-off.
>
> I am excited about the choice of the problem, but I have reservations about the treatment of privacy in the paper. First, KL divergence is a very weak (average case) notion privacy that can be easily broken. Second, the algorithm that is outlined in the paper gives an empirical way to compute the representation while balancing the utility-privacy trade-off (Eq. 6). However, there is no formal privacy guarantee for the algorithm. It is important to remember that unlike the utility, privacy is a worst-case notion and should formally hold in all occasions.’
> #
>
> *RESPONSE:
> Since this work deals with data-driven privacy, it is not possible to know beforehand the exact model used to generate the observed data, this is a common occurrence in real scenarios, which, in our opinion, makes it an interesting problem (we nevertheless added results on data with known distributions in the revised version). Under those constraints, it is challenging to provide guarantees similar to the ones made by differential privacy, this work is an initial step in that direction and something we want to pursue in the future.
>
> The reviewer correctly observes that this framework provides privacy in expectation, we are currently investigating how the variance in this type of privacy can be measured and bounded.
>
> Finally, we also agree that privacy is a worst-case notion, which is why the secret inference algorithm is trained adversarially, to test whether any possible attacker can still learn anything meaningful from the sanitized representation.
>
> These comments have now been incorporated into the revised manuscript. Clarifications on the adversarial issues were added in paragraph 1 in Experiments and Results (page 6), Experiments on known distributions are shown in Supplementary Material (Section 7.2), and comments were added to Concluding remarks reflecting the comments above.
>
> To conclude, we have addressed all the reviewer’s comments and we hope he/she will now support accepting this paper, reflecting the statement “I am excited about the choice of the problem.” While as with most papers there is still significant work to be done, the paper proposes a new important framework for privacy with new results and theory, as stated by the reviewer as well.

---

### Official Review · AnonReviewer3 · 2018-11-05
**Nice idea. Need more clarification.**

**Rating:** 6
**Confidence:** 4

**Review:**

This paper proposes a privacy framework where a privatizer, according to the utility and secret specified by users, provides a sanitized version of the user data which lies in the same space as the original data, such that a utility provider can run the exact algorithm it uses for unsanitized data on the sanitized data to provide utility without sacrificing user privacy. The paper shows an information theoretic bound on the privacy loss and derives a loss function for the privatizer to use. It then proposes an algorithm for the privatizer, evaluates its performance on three scenarios.

The paper investigated on an interesting problem and proposed a nice solution for synthetic data generation. However, I think the proposed framework and how the example scenarios fit into the framework needs to be described clearer. And more experimental evaluations would also help make the result more solid.

More detailed comments:
- Do the user and privatizer need to know what the machine learning task is when doing the sanitization? Is it ok for the privatizer to define utility in a different way as the machine learning task? For example, as a user, I may want to hide my emotion, but I’m ok with publishing my gender and age. In this case, can I use a privatizer which defines secret as gender and utility as (gender, age)? And will the synthetic data generated by such a privatizer be equally useful for a gender classifier (or an age classifier)? It would be good if it is, as we don’t need to generate task-specific synthetic data then.
- I think it might be interesting to see the effect of the privatizer when utility and secrecy are correlated (with a potentially different level of correlation).
- It’s not clear to me where the privatizer comes into the picture in the subject-within-subject example. It seems like users here are people whose face appear in front of the mobile device, so they probably won’t be able to privatize their face image, yet the device won’t be able to tell if users are in the consenting group without looking at their faces. I think it’s better if more clarification on how each of the three scenarios fits into the proposed framework is provided.
- Can different user have different secret?
- In the experiment, it might be better to try different models/algorithms for the utility and secrecy inferring algorithm, to demonstrate how the privatizer protects secrecy under different scenarios.
- I think there might be some related work on the field of fairness and transparency where we sometimes want the machine learning models to learn without looking at some sensitive features. It would be nice to add more related work on that side.
- It’s better to give more intuition and explanation than formulas in Section 3.
- There are a few typos (e.g. Page2, 3rd paragraph, last sentence: “out”-> “our”; Equation (4), I(S, Q) should be I(S; Q)?; Page 8, 2nd paragraph, 1st line “Figures in 5” -> “Figure 5”) that need to be addressed. Texts in some figures, like Figure 2 and 3, might be enlarged.

---

> ### Author Response · Authors · 2018-11-23
> **Response to Reviewer**
>
> We are very excited about the positive and enthusiastic support of all the reviewers, and their outstanding feedback. We thank the reviewers for the constructive comments. We have taken all the reviewers' points into consideration and have modified the paper accordingly, see blue text in the revised document for the major changes (all related to clarifications, following the ICLR policy). A detailed point-by-point answer is provided below. Overall, the paper was revised for clarity, adding additional intuition to the theory and applicability of the framework. We also clarified the purpose of some of the experiments and added material in the Supplementary section to show additional examples on controlled situations (to further explain the bounds and their relevance), as well as extensive implementation details.
>
> We modified the first paragraph of the Experiments and results section to better motivate the scenarios shown, the third paragraph on page 7 was also modified to better motivate the subject-within-subject example (a critical application for mobile devices and smart speakers for example). The first two paragraphs of section 4 on page 5 were modified to better motivate the proposed loss.  Section 7.2 in Supplementary material added experiments under controlled conditions to better analyze the performance of the algorithm in a dataset where the underlying distributions are known. Section 7.3 (see supplementary) shows a detailed overview of the training algorithms and architectures used throughout the paper. All these additions resulted from the excellent feedback from the reviewers.
>
> #REVIEWER QUOTE:
> ‘More detailed comments:
> Do the user and privatizer need to know what the machine learning task is when doing the sanitization? Is it ok for the privatizer to define utility in a different way as the machine learning task? For example, as a user, I may want to hide my emotion, but I’m ok with publishing my gender and age. In this case, can I use a privatizer which defines secret as gender and utility as (gender, age)? And will the synthetic data generated by such a privatizer be equally useful for a gender classifier (or an age classifier)? It would be good if it is, as we don’t need to generate task-specific synthetic data then.’
> #
>
> *RESPONSE:
> The abstract and paragraphs 2 and 6 in the Introduction were modified to clarify this. The privatizer does not necessarily need to know the utility inference mechanism when performing sanitization. It is, however, extremely advantageous to know the algorithm used, this allows the privatizer and service provider to collaborate, this is what enables scalability in the way you describe (multiple privacy tasks served by a single utility algorithm)
>
> #REVIEWER QUOTE:
> 'I think it might be interesting to see the effect of the privatizer when utility and secrecy are correlated (with a potentially different level of correlation). '
> #
>
> *RESPONSE:
> Section 7.3 was added to show this behavior on data with known properties and distributions. Importantly, it shows that the privacy learning mechanism effectively approaches the bounds regardless of correlation levels on this data. We should also add that the examples here introduced are challenging in the sense that privacy is often a significantly easier task than the utility (e.g., gender detection is easier than person identification).
>
> #REVIEWER QUOTE:
> 'It’s not clear to me where the privatizer comes into the picture in the subject-within-subject example. It seems like users here are people whose face appear in front of the mobile device, so they probably won’t be able to privatize their face image, yet the device won’t be able to tell if users are in the consenting group without looking at their faces. I think it’s better if more clarification on how each of the three scenarios fits into the proposed framework is provided.'
> #
>
> *RESPONSE:
> The subject-within-subject example was meant to illustrate that a filter such as this, applied as close as possible to the sensor level, can essentially provide assurances that non-consenting subjects that stand close to the phone would have their privacy preserved. This would be a two-stage process where the image is first sanitized in a trusted environment (a closed box that can be certified to not disclose anything other than the sanitized image), and then the sanitized image is disclosed to the utility-provider, in this case, the phone-unlocking app. Paragraph 3 in page 7 was modified to reflect this. Here “privacy” is not an attribute of a subject but the subject him/herself. Same for utility. Also, note that this example illustrates how the same theoretical and computational framework can address very different problems.

---

> > ### Author Response · Authors · 2018-11-23
> > **Response to reviewer continued**
> >
> > #REVIEWER QUOTE:
> > 'Can different user have different secret? '
> > #
> >
> > *RESPONSE:
> > Yes, different users may define different secrets, a key concept of this work is to have users in control of their privacy desires and needs. This is made scalable by ensuring the sanitized images work on the existing utility-providing networks (a many-to-one relationship between secrecy preferences and utility); this was clarified in the abstract and paragraphs 2 and 6 in the Introduction.
> >
> > #REVIEWER QUOTE:
> > 'In the experiment, it might be better to try different models/algorithms for the utility and secrecy inferring algorithm, to demonstrate how the privatizer protects secrecy under different scenarios.'
> > #
> >
> > *RESPONSE:
> > We agree that the results could be strengthened by testing against various utility-inference algorithms. We did, however, train against secrecy adversaries that were constantly adapting to the sanitation strategy, this provides some guarantee that a yet-unobserved secrecy inferring algorithm cannot violate privacy, especially since the secrecy adversaries came from a sufficiently rich parametric family (DNN).
> >
> > #REVIEWER QUOTE:
> > 'I think there might be some related work on the field of fairness and transparency where we sometimes want the machine learning models to learn without looking at some sensitive features. It would be nice to add more related work on that side. '
> > #
> >
> > *RESPONSE:
> > We are looking into that as a future research direction. Indeed be believe these concepts are related (though note that a system doesn’t need to be private to be fair). We also believe that a unified theory of privacy, fairness, and transparency will be a superb contribution to the community. We have added a comment on this in the revised version.
> >
> > #REVIEWER QUOTE:
> > 'It’s better to give more intuition and explanation than formulas in Section 3.'
> > #
> >
> > *RESPONSE:
> > Paragraph 1 and 2 in Section 4 were modified to give a better summary of the main ideas behind the loss functions. Note also that we added additional experiments with known data distributions to further stress the loss functions and the bounds.
> >
> > #REVIEWER QUOTE:
> > 'There are a few typos (e.g. Page2, 3rd paragraph, last sentence: “out”-> “our”; Equation (4), I(S, Q) should be I(S; Q)?; Page 8, 2nd paragraph, 1st line “Figures in 5” -> “Figure 5”) that need to be addressed. Texts in some figures, like Figure 2 and 3, might be enlarged.'
> > #
> >
> > *RESPONSE:
> > Typos have been addressed. Thanks.
> >
> > To conclude, we have addressed all the reviewer’s comments and we hope he/she will further support accepting this paper. While as with most papers there is still significant work to be done, the paper proposes a new important framework for privacy with new results and theory.

---

### Meta-Review · Area_Chair1 · 2018-12-17
**Misses the point of privacy**

**Confidence:** 5
**Recommendation:** Reject

**Metareview:**

This paper addresses data sanitization, using a KL-divergence-based notion of privacy. While an interesting goal, the use of average-case as opposed to worst-case privacy misses the point of privacy guarantees, which must protect all individuals. (Otherwise, individuals with truly anomalous private values may be the only ones who opt for the highest levels of privacy, yet this situation will itself leak some information about their private values).